# Deep learning and transfer learning identify breast cancer survival subtypes from single-cell imaging data

Shashank Yadav[1,2], Shu Zhou[1,2], Bing He[1], Yuheng Du[1] & Lana X. Garmire [1✉]

## Abstract

**Background** Single-cell multiplex imaging data have provided new insights into disease subtypes and prognoses recently. However, quantitative models that explicitly capture single-cell resolution cell-cell interaction features to predict patient survival at a population scale are currently missing.

**Methods** We quantified hundreds of single-cell resolution cell-cell interaction features through neighborhood calculation, in addition to cellular phenotypes. We applied these features to a neural-network-based Cox-nnet survival model to identify survival-associated features. We used non-negative matrix factorization (NMF) to identify patient survival subtypes. We identified atypical subpopulations of triple-negative breast cancer (TNBC) patients with moderate prognosis and Luminal A patients with poor prognosis and validated these subpopulations by label transferring using the UNION-COM method.

**Results** The neural-network-based Cox-nnet survival model using all cellular phenotype and cell-cell interaction features is highly predictive of patient survival in the test data (Concordance Index > 0.8). We identify seven survival subtypes using the top survival features, presenting distinct profiles of epithelial, immune, and fibroblast cells and their interactions. We reveal atypical subpopulations of TNBC patients with moderate prognosis (marked by *GATA3* over-expression) and Luminal A patients with poor prognosis (marked by *KRT6* and *ACTA2* over-expression and *CDH1* under-expression). These atypical subpopulations are validated in TCGA-BRCA and METABRIC datasets.

**Conclusions** This work provides an approach to bridge single-cell level information toward population-level survival prediction.

## Plain language summary

It may be possible to separate patients with cancer into different groups or subtypes based on the features of their tumor, such as the interactions between different types of cells in the tumor. In this study, we develop a computer-based model to calculate the interactions between cells in breast cancer images. We use these interactions to identify seven subtypes of patients with breast cancer with differences in their survival. We identify some subpopulations of patients with atypical survival outcomes. This work may ultimately help clinicians and researchers to identify patients with breast cancer at increased risk of poorer outcomes and to tailor their treatments accordingly.

[1] Department of Computational Medicine and Bioinformatics, University of Michigan, Michigan, MI 48105, USA. [2]These authors contributed equally: Shashank Yadav, Shu Zhou. ✉email: lgarmire@med.umich.edu

Breast cancer became the most commonly diagnosed cancer in 2020, with an estimated 2.3 million new cases globally[1]. Predicting survival in breast cancer can aid clinicians in making prompt prognostic decisions and deciding the direction of treatment. The relevance of prognosis in oncology is projected to grow in the future as new prognostic indicators allow for more precise treatment therapies[2]. However, insufficient knowledge about the intercellular interaction between the tumor and tumor microenvironment is a big roadblock in applying personalized therapies. The breast cancer tumor ecosystems in breast cancer consist of neoplastic epithelial cells forming the tumor core, as the 'tumor microenvironment' is composed of several types of immune cells, fibroblasts, adipocytes, and mesenchymal cells[3,4]. These diverse cell types alter molecular and cellular programs and present dynamic spatial heterogeneity as the disease progresses. Such temporal and spatial changes are responsible for differential responses to anti-cancer therapies and subsequent clinical outcomes. Hence, it is essential to develop a comprehensive understanding of breast cancer heterogeneity by elucidating the contribution of tumor, tumor microenvironment, and their intricate interactions[5,6].

Investigations at single-cell resolution[7,8] have lately enabled detailed elucidation of tumor-microenvironment interactions. Various components of tumor-immune-stromal relationships have been identified, and tumor heterogeneity was also analyzed to distinguish breast cancer subtypes based on epithelial and immune cell populations[9–11]. Potential biomarkers have also been discovered for personalized cancer immunotherapy through Single-cell RNA sequencing (scRNA-seq)[12]. However, one major limitation of scRNA-seq is the loss of spatial information crucial in understanding tumor heterogeneity in situ[13]. To address these issues, spatially resolved assays, such as single-cell transcriptomic and proteomic techniques, have been developed to study the distribution of cells in cancers[14]. Recently, Imaging Mass Cytometry (IMC)-based approaches have quantified tumor heterogeneity with spatial context and identified new breast cancer molecular subtypes in large population cohorts[15,16]. However, most of these studies that detect molecular subtypes are unsupervised, without explicitly fitting phenotypes such as survival in the learning process. Rather, survival is used as a post hoc metric to evaluate the subtypes[17]. Moreover, these unsupervised approaches cannot be directly used to predict uncovered patients, limiting the practical utility of the subtype findings.

In this study, we asked whether quantifying single-cell level cell-cell interactions could provide meaningful insights for prognosis prediction for breast cancer patients. We computed single-cell level imaging features that capture cell-cell interactions in breast cancer tissues from a previous single-cell imaging mass cytometry study on 259 breast cancer patients with survival data[16]. We applied a neural network-based method Cox-nnet previously developed in our group[18] on these features to predict explicitly the patient survival outcome. Using the top survival features, we uncovered seven single-cell interaction-based patient subtypes that show better associations with survival compared to the current mainstream molecular subtypes in breast cancer. We characterized these survival subtypes with distinct profiles of cellular phenotypes as well as cell-cell interactions. Using this survival subtyping classification, we identified two atypical patient subpopulations: a subgroup of triple-negative breast cancer patients with moderate prognosis and a subgroup of Luminal A patients with poor prognosis. We further utilized the transfer learning approach and validated the presence of such unconventional subgroups and their biomarkers in the TCGA and METABRIC breast cancer datasets.

## Methods

**Dataset and extracted features.** The dataset analyzed in the study is obtained from a previously published study containing 259 patients[16]. The dataset contains the IMC data, phenograph neighborhood information, clinical features (Tumor Grade, ER Status, PR Status and HER2 Status), and patient prognosis outcome (overall survival time, disease-free survival time, and alive/dead status). The previous work defined 27 cellular phenotypes that described the histopathological landscape of breast cancer, and direct neighbor cells are defined within 4 pixels (4 μm) distance to the target cell. In this study, we used the clinical features, cellular phenotype density (count of cells of each phenotype per unit area), and cellular neighborhood information for analysis, as detailed below:

*Clinical features.* The clinical features set comprises the Tumor Grade, ER Status, PR Status, and HER2 Status for each patient. Tumors in which cells appear highly dissimilar to normal cells tend to proliferate, and the tumor grade is assigned based on the extent of proliferation. In the traditional clinicopathological classification of breast cancer, patients are classified into Luminal A, Luminal B, Triple-Negative, and HE2-Enriched classes. This classification is based on patients' clinical features such as ER (positive/negative), PR (positive/negative), and HER2 (positive/negative) status[19,20].

*Cellular phenotypic features.* For each patient, the cell phenotype density is quantified as the counts of each cellular phenotype per unit area in the IMC image of the tumor tissue. Out of the original 27 cellular phenotypes, there are six immune, seven stromal, and fourteen epithelial cellular phenotypes. The six immune phenotypes are B Cell, T and B Cell, $T-Cell_1$, $Macrophage_1$, $T-Cell_2$, $Macrophage_2$. The seven stromal phenotypes are Endothelial, $Vimentin^{hi}$ Fibroblasts, Small circular Fibroblasts, Small elongated Fibroblasts, $Fibronectin^{hi}$ Fibroblasts, Large Elongated Fibroblasts, $SMA^{hi}$-$Vimentin^{hi}$ Fibroblasts. The 14 epithelial cellular phenotypes are Hypoxic Epithelial Cells, Apoptotic Epithelial Cells, Proliferative Epithelial Cells, $p53^+$ $EGFR^+$ Epithelial Cells, Basal CK Epithelial Cells, $CK7^+CK^{hi}Cadherin^{hi}$ Epithelial Cells, $CK7^+CK^+$ Epithelial Cells, $Epithelial^{low}$ Epithelial Cells, $CK^{low}HR^{low}$ Epithelial Cells, $CK^+HR^{hi}$ Epithelial Cells, $CK^+HR^+$ Epithelial Cells, $CK^+HR^{low}$ Epithelial Cells, $CK^{low}HR^{hi}p53^+$ Epithelial Cells, Myoepithelial Cells.

*Cell-cell interaction features.* The phenotypic features in the original report are limited, as they do not assess the interactions between cellular phenotypes and between tumor-tumor microenvironments, which are important parts of tissue heterogeneity. We utilized the available phenograph[21] neighborhood-information data from each patient and quantified the binary interactions between cellular phenotypes. The phenograph neighborhood of the IMC image is described as a numerous cellular network spread out in the mass cytometry image, where individual cells are represented as nodes of the cellular network. Starting with 27 cellular phenotypes, we calculated 378 pairwise phenotype-phenotype features.

Cell-cell interactions are restricted to those within the same cellular community. We iterate through each cell of a cellular community in the image, enumerate the binary interactions the cell makes with its neighbors, and multiply the sum of the interactions by the clustering coefficient $C$ of the cellular community as the weight. Then, we repeat this process for all the cellular communities across the mass cytometry image, sum the 378 interactions, and divide them by the area occupied by all

the cells. We define the result of this procedure as the 'Cell-Cell Interaction Score' (CCIS).

$$CCIS_{\forall Pair(x,y), x \in P, y \in P} = \frac{\sum_{\forall N}(C * B_{xy})}{A} \qquad (1)$$

where $C$ is the clustering coefficient for each cluster, $B_{xy}$ is the binary interaction counts between a particular pair of phenotypes $x$ and $y$, $A$ is the total area occupied by all the cells in a mass cytometry image, $P$ is the set of all phenotypes and $N$ is the number of cellular communities in an image.

At a more detailed level, these 378 can be further categorized by immune/stromal/epithelial cell types in the interacting pairs and classified as immune-immune (21 features), immune-stromal (42 features), immune-epithelial (84 features), stromal-stromal (28 features), stromal-epithelial (98 features), epithelial-epithelial (105 features) interactions. Among the feature pairs, 1 immune-stromal feature (macrophage interacting with Large Elongated Fibroblasts), 2 immune-epithelial features (macrophage interacting with Hypoxic Epithelial Cells and macrophage interacting with CK7$^+$CK$^{hi}$Cadherin$^{hi}$ Epithelial Cells), 1 stromal-stromal feature (Large Elongated Fibroblasts self-interaction) and 1 stromal-epithelial (Large Elongated Fibroblasts interacting with Myoepithelial cells) are dropped since these pairwise phenotype-phenotype interactions calculations returned 0 (lack of neighbors within the immediate vicinity), hence the final number of features would be 373 features. We plot the feature heatmaps using the 'heatmap' package in R.

**Survival modeling.** We use a variety of neural-network-based Cox-nnet models as the one-stage Cox-nnet-v2 models for the phenotypic feature set, tumor-microenvironment feature set, tumor-core feature set, and pairwise combinations. Further, we develop a two-stage Cox-nnet model by combining the hidden layer output of previously trained one-stage models. For each one-stage Cox-nnet-v2 model, the hidden layer nodes are equal to the square root (rounded up) of input nodes. The Cellular Phenotypic (CP), tumor-microenvironment interaction (TMI), and tumor-core interaction (TCI) feature-based models have 27, 268, and 105 input features, respectively, and hence we obtain 6, 17, and 11 nodes in the respective hidden layer. The two-stage Cox-nnet model combines the hidden layer output features. As a result, 34 input features are used in the second stage. For comparison, we use the Cox-PH model on clinical features as the baseline model.

**Survival feature ranking.** We calculate the feature importance of the 34 input features using the 'Variable Importance' function of the Cox-nnet-v2 model. Out of the 34 input features in the two-stage Cox-nnet model, we calculate the importance scores, separate them into the original three sets of 6, 17, and 11 features, respectively, and associate them with the one-stage Cox-nnet-v2 models. For each model, we took a dot product of the hidden layer feature-importance vector with the model weights to get the importance scores vector of the original features.

$$\hat{O} = W^T.\hat{H}, \qquad (2)$$

where $\hat{O}$ is the importance vector of the original features (shape1). $W$ is the model weight matrix (shape $h \times p$). $H$ is the importance vector for hidden layer (shape $h \times 1$).

**Unsupervised analysis for patient subtype detection.** We normalize the feature importance values between 0 and 1 and select the top features from each of the three feature sets. For the phenotypic set, we set the threshold as 0.5 and selected all the features with a higher importance score. For the

tumor-microenvironment and tumor-core feature sets, we select features with an importance score greater than 0.75. In total, out of the 40 features we choose 50 features and perform NMF-based consensus clustering using the NMF R package v0.23.0. This technique has been used in molecular subtype detection in cancer[22,23]. We carry out a hyperparameter search for the NMF rank and varied it from 3 to 15, finding that the maximum values of 'cophenetic scores' and 'silhouette coefficients' are reached at an NMF rank of seven, and hence we choose the optimum NMF rank as seven for subsequent analysis.

**Feature correlation analysis and comparison between NMF-defined and clinically defined subtypes.** We calculate the correlations between the features associated with each NMF-defined subtype using Spearman's Correlation Coefficient in R and plot the Circos plot of the correlation using the circlize package in R[24]. How the NMF-defined classes intersect with the clinicopathological classification is determined by a Sankey Plot. We plot the Sankey plot using the plotly package in Python.

**Class label transfer and comparison with TCGA-BRCA and METABRIC datasets.** We verify our results for 'TNBC-Good Survival' and 'Luminal A-Poor Survival' patient subpopulations in two external datasets, TCGA-BRCA and METABRIC[25,26]. Here, we utilize the mass cytometry counts for 30 protein-based biomarkers and calculate the average expression of each biomarker over all the cells present in the IMC image. We treat it as pseudo-bulk protein expression and a proxy for bulk mRNA expression to facilitate the comparison with bulk mRNA expression-based external datasets. Then, we repurpose the UNION-COM algorithm[27] to perform patient matching between our dataset, the TCGA-BRCA dataset, and the METABRIC dataset. The UNION-COM method was originally developed to perform topological alignment and label transfer on single-cell multi-omics datasets. It takes two expression matrices as input, calculates the joint embedding between the two datasets, and maps samples from one dataset to another. In the TCGA-BRCA dataset, we find 28 genes corresponding to 28 protein-based biomarkers out of 30 (except TWIST and mTOR). In the METABRIC dataset, we find all 30 genes corresponding to the 30 protein-based biomarkers. For each TNBC and Luminal A sub-population, the pseudo-bulk protein expression matrix from the single-cell dataset is used as input, and the mRNA expression matrix is applied as the external dataset in the UNION-COM method. We run the UNION-COM method using default parameters, and it returns the pairwise distance between patients between our dataset and the external dataset. We set a 99.5% similarity cutoff and select patients matching our 'TNBC-Good Survival' and 'Luminal A-Poor Survival' patient subpopulations, respectively. Survival plots are made using the lifelines Python package, and differential gene expressions are done using the limma package in R[28].

**Cross-check the TNBC survival subtypes with the molecular subtypes of TNBC.** We cross-compared the seven subtypes in the previous study[29] with the two survival subtypes we identified. We used the six subtype markers (MKI67, MYC, EGFR, CDH1, SNAI2 and TWIST1) in that study that overlap with the 30 protein-based biomarkers in the single-cell imaging dataset here. The six markers are MKI67 and MYC as the proliferation markers, EGFR as the growth factor/Myoepithelial markers, and SNAI2 and TWIST1 as the epithelial-mesenchymal transition–associated (EMT-associated) markers. The raw data from the previous study are not available; we relied on the pixels in this heatmap as the normalized gene expression values for the

comparative study. We first transformed the heatmap of Supplementary Fig. 6[29] of the six markers from RGB (red, green and blue) to HSV(Hue, Saturation and Value) scale and filtered out all the red and green-related colors by setting the HSV thresholds. Then we transformed the HSV scale to the gray scale and mapped the intensity to log-transformed gene values according to the figure legend. Specifically, we map green-related colors from −2 to 0 and red-related colors from 0 to 2. We plotted the level of the six markers from the imaging data alongside the heatmap from the previous study. We performed hierarchical clustering with the average expression level of markers from the two survival subtypes defined by us and the TNBC molecular subtypes obtained from the previous study.

## Results

**Feature engineering from the breast cancer single-cell images.** The breast cancer cohort contains 259 patients with survival data, as reported earlier[16]. The summary of the cohort's patients, including clinical variables such as tumor grades, clinical features (ER, PR, HER2), and clinicopathological classes, is shown in Supplementary Table 1. We first extracted 27 pre-defined cellular-phenotype (CP) features based on image mass cytometry data (Fig. 1a, Supplementary Data 1). These features describe epithelial, immune, and stromal cell phenotypes at the single-cell resolution in the original study ("Methods"). We next used networks of cells to represent the cells and cellular communities with the spatial arrangement as shown in the imaging data. We then calculated phenograph neighborhood information based on the cellular phenotypes in the tissue ("Methods"). This process results in an additional 378 cell-cell interaction features that can be broadly divided into two subsets. The first subset contains 273 features related to the tumor-microenvironment interaction (TMI) features (Fig. 1a). The TMI features are computed from pairwise interactions among the three types of cells: immune, stromal, and epithelial cells. Specifically, they represent immune-immune (21 features), immune-stromal (42 features), immune-epithelial (84 features), stromal-stromal (28 features), and stromal-epithelial (98 features) interactions. The second subset contains 105 tumor-core interaction (TCI) features, which are exclusively epithelial-epithelial interactions between epithelial cells of various cellular phenotypes (Fig. 1a). The distributions of these 378 features in patients together with clinically defined breast cancer subtypes are illustrated in Fig. 1b. Anchoring on the TCI features, hierarchical clustering demonstrates broad but distinct cellular heterogeneity among patients, far more complex than that defined by the clinical subtypes. Compared to TCI features, TMI features describing immune-epithelial and stromal-epithelial interactions have similar but less distinct heterogeneity. On the other hand, all CP features show far less global correlations with those TCI patterns presented in epithelial-epithelial interactions.

**Survival prediction using single-cell phenotype features.** A major goal of our study is to identify single-cell level features associated with patient survival and further evaluate the relative contributions of these features toward patient survival. To this end, we used the recently developed Cox-nnet neural-network-based survival prediction models from our group (Ching et al.[18]), which had shown advantages in single or multiple data modalities[30,31], compared to the conventional Cox-PH method[32]. Here as a comparison to Cox-nnet models, we built a Cox-PH model using the clinical information including ER status, PR status, HER2 status, and tumor grade and TNM (tumor-node-metastasis) stagings (Set I) as the baseline model. We constructed a series of two-stage Cox-nnet models (Fig. 2a) on the

combinations (Set II, III, IV, and V) of CP features, TMI features and TCI features (Set II, III, IV, and V). Two-stage Cox-nnet models are complex models that use simpler Cox-nnet models as individual building blocks[30] to integrate different feature sets or data modalities. In the first stage of training, each Cox-nnet model is built to fit a specific set of data (CP, TMI or TCI features) to predict survival. The hidden nodes in the first-stage Cox-nnet model are then combined as the input features to train a second-stage Cox-nnet model. For each Cox-nnet model, we used L2 regularization to reduce overfitting.

To rank the relative contributions of different features, we estimated the relative importance of the features in each of the CP, TMI, and TCI feature sets (Supplementary Data 2–4). Among CP features, immune cells, including type 2 macrophages (CD68+/vimentin low) and T & B immune cell clusters, have the highest relative importance scores (1.000 and 0.692 respectively) compared to those (0.507–0.631) of different subtypes of epithelial cells (Supplementary Data 2), highlighting their importance in patient prognosis. In TMI features, two types of immune-epithelial interactions (T and B cells–CK$^{low}$HR$^{low}$ epithelial cells, Macrophage$_1$–CK$^{low}$ HR$^{hi}$ p53 $^+$) and a type of fibroblast–epithelial interaction (small elongated fibroblast–CK7$^+$CK $^+$ epithelial cells) and are top 3 dominant features (Supplementary Data 3). Among the TCI features, the interactions among most proliferative epithelial cells are the strongest, as expected (Supplementary Data 4).

**Identifying survival subtypes among breast cancer patients.** Our next goal was to identify patient subpopulations associated with survival, using the top survival features selected by importance scores from the best model using Set XII (the combination of CP, TMI, and TCI features). We performed Non-negative Matrix Factorization (NMF) based consensus clustering on the top 50 features according to the importance scores ("Methods"). As a result, we identified seven optimal patient subpopulation clusters, indexed by 1–7 from the best to the worst survival risks (Fig. 3a). We confirmed that seven clusters are the optimum value, based on Cophenetic score and Silhouette coefficient, two metrics of clustering accuracy (Fig. 3b). The Kaplan-Meier plot for the overall survival of the seven survival subtypes yields a much higher C-index of 0.80 and a more significant log-rank $p$-value $p = 8.6e^{-0.6}$ (Fig. 3c) for $n = 259$ independent patients, compared to the C-index of 0.63 and the log-rank $p$-value of 0.001 from the stratification of the four molecular subtypes (Fig. 3d). These results demonstrate that the single-cell level CP, TMI, and TCI features yield more informative subtypes that better reveal the heterogeneity in survivorship among patients.

We further describe the survival subtypes based on their top features and their relationships with other clinical information (Fig. 3e, Supplementary Table 2). As expected, the better survival subtypes tend to have more ER+ and PR+ cases, and the worst survival subtypes tend to have higher tumor grade (Grade III) and HER2- cases. The best survival subtype 1 is enriched with a subtype of epithelial cells with high levels of cytokeratin (CK) 7 and 19 but low hormone receptors (CK7, CK19, low HR). It also has high levels of interaction scores between this epithelial cell subtype and several subtypes of fibroblast cells, highlighting the importance of the interaction between these two cell types for patient outcomes. The next best subtype 2 also has enriched scores on a subtype of epithelial cells that express pan-cytokeratins and hormone receptors (pan-CK, HR), as well as their interactions with certain fibroblast cells. On the other hand, the subtype 7 with the worst survival is characterized by high degrees of interactions among proliferative epithelial cells, as well as interactions between macrophages/T-cells expressing high

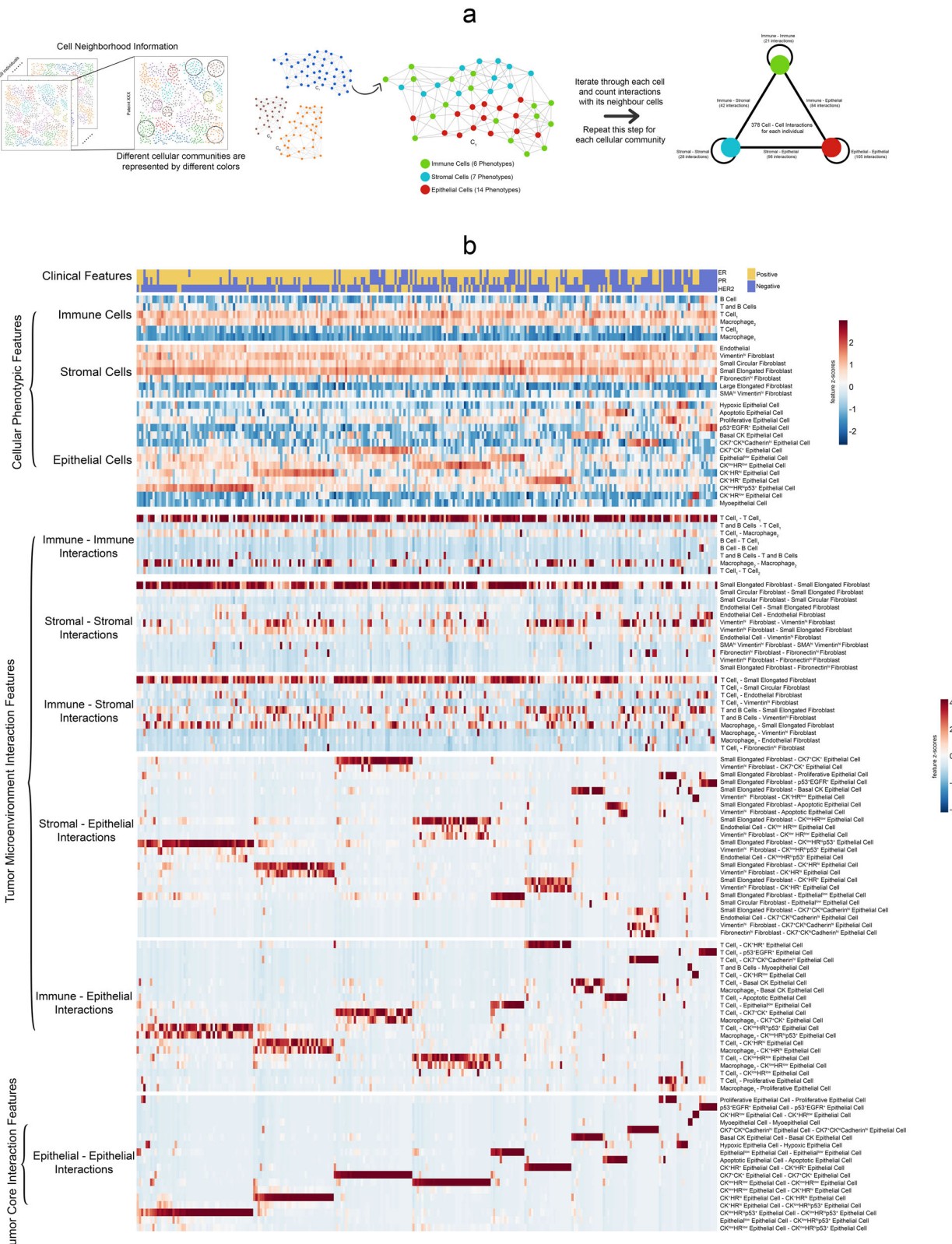

**Fig. 1 Overview of different feature sets in the data. a** Methodology for extracting the cell-cell interaction features. We utilize the cellular neighborhood information obtained from the phenograph. The phenograph result contains different locally connected cellular communities. For each cellular community (the neighborhood graph), we iterate through each cell (node) and count the number of interactions between the particular cell and its neighbors. We repeat this process for all the different cellular communities to assess the 378 different pairwise cell-cell interactions. **b** Heatmaps illustrate the different feature types for all the patients. The patients are arranged in the order defined by the hierarchical clustering based on the epithelial-epithelial interactions.

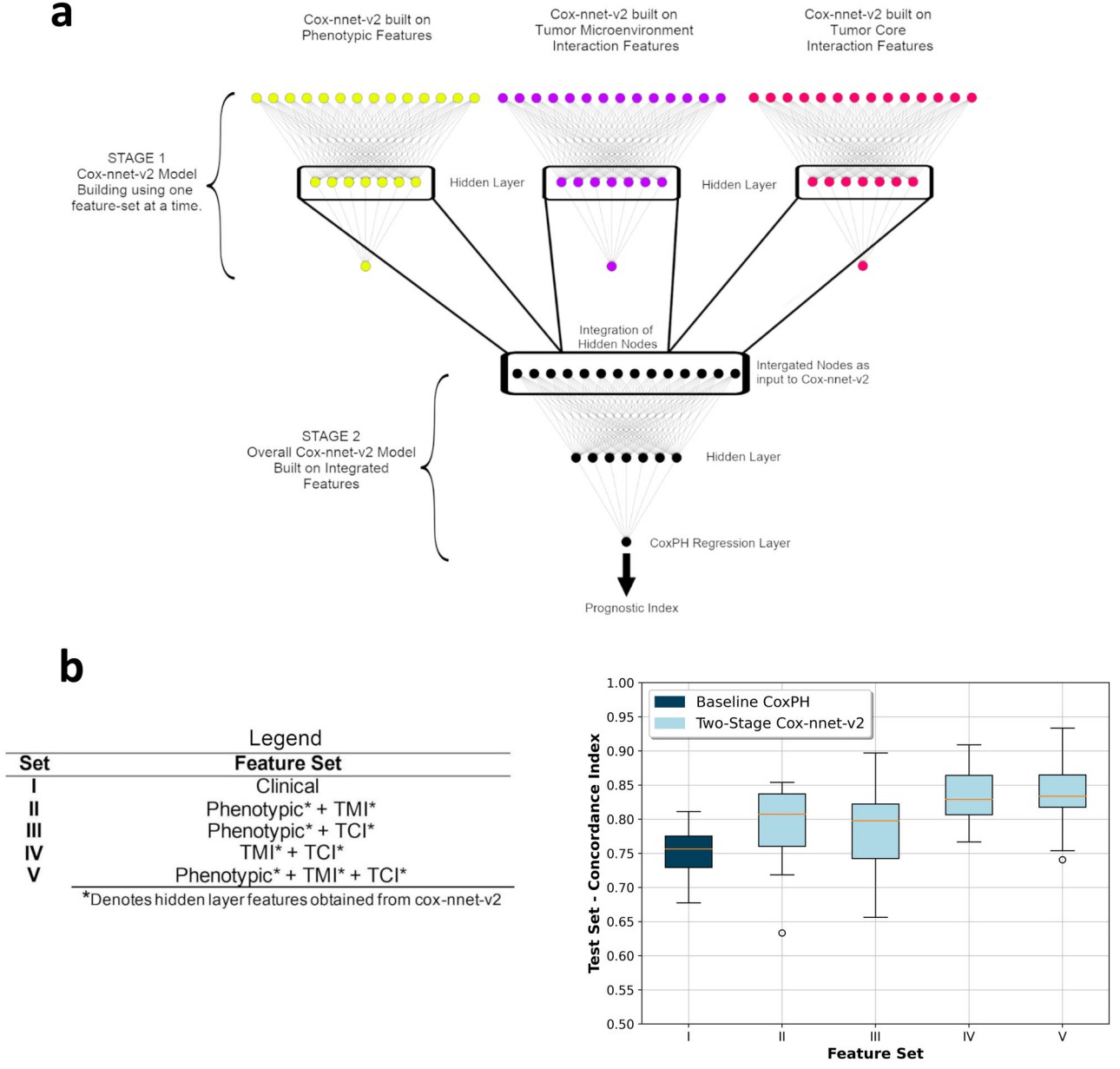

**Fig. 2 Two-stage Cox-nnet model comparison with baseline and Cox-nnet-v2. a** Model architecture for the two-stage Cox-nnet model. Three individual Cox-nnet models were built for each data type. The hidden nodes from the first-stage Cox-nnet models were combined to form the input to construct a Cox-nnet model in the second stage. **b** Comparison of Concordance index across different Feature Set-Model pairs through boxplot with $n = 20$ independent experiments. The box displays the quartiles of the dataset, and the whiskers show the rest of the distribution, except for points that are determined to be outliers, using a method that is a function of the inter-quartile range.

vimentin and the proliferative epithelial cells. The second worst survival subtype 6 has a high level of epithelial cells lacking CK and HR expression, as well as strong interactions between fibroblast/B cells and these epithelial cells. In summary, the survival subtypes show distinct profiles of epithelial cell subtypes and interactions between the epithelial cells and adjacent immune and fibroblast cell subtypes.

We also compared the seven survival clusters with the 18 single-cell pathology (SCP) subgroups based on the 27 cellular phenotype features, as identified from the original study[16]. As shown in Supplementary Fig. 1, cluster 1 has a strong association (90%) with SCP_11 (CK7 and pan-CK) subgroup. Cluster 5 is enriched with SCP_17 (Hypoxic) subgroup. Cluster 6 is equally distributed in SCP_7 (Epithelial.low) and SCP_10 (Epithelial.low mixed) subgroups. Cluster 7 shows a connection with the SCP_14

(Proliferative) subgroup. Cluster 2, Cluster 3 and Cluster 4 do not show a clear association with some particular SCP subgroups.

**Characterization of the seven survival subtypes**. We next directly compared the enrichment of different types of immune, fibroblast and epithelial cells, as well as their interaction scores in the seven survival subtypes (Fig. 4). Subtypes 5–7 show quite distinct patterns from subtypes 1–4. In particular, subtype 5 has a high level of hypoxic epithelial cells (Fig. 4e), as well as high levels of interactions between these hypoxic epithelial cells and macrophage₂, T/B cells and vimentin-expressing fibroblasts (Fig. 4i, k and o). Subtype 6 has the second highest level of proliferative epithelial cells (Fig. 4f) but the lowest levels of vimentin expression fibroblast cells (Fig. 4c), and accordingly, the second highest

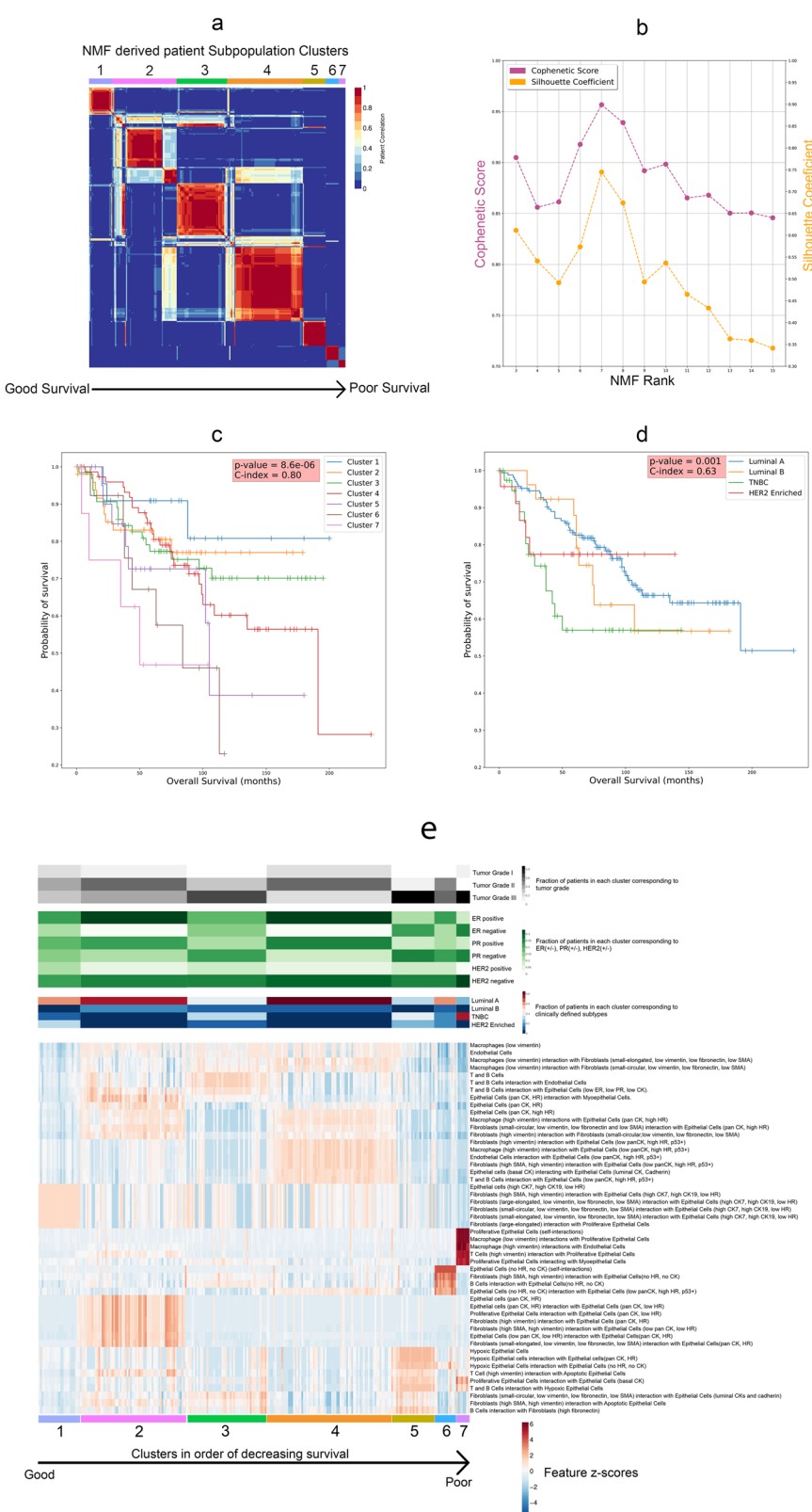

**Fig. 3 Non-negative Matrix Factorization (NMF)-based subpopulation detection associated with survival. a** NMF heatmap for 259 patients in our cohort illustrating the seven subpopulations arranged in order of decreasing survival. **b** Cophenetic Score and Silhouette Coefficient versus the NMF Rank. **c** Kaplan-Meier (KM) plots illustrate the Overall Survival for patients in the seven NMF-derived subpopulation clusters. **d** Kaplan-Meier plots illustrate the Overall Survival for patients based on clinicopathological classification. **e** Heatmap of top-ranked features from best performing two-stage Cox-nnet model, in associations with tumor grade, clinical features and molecular subtypes.

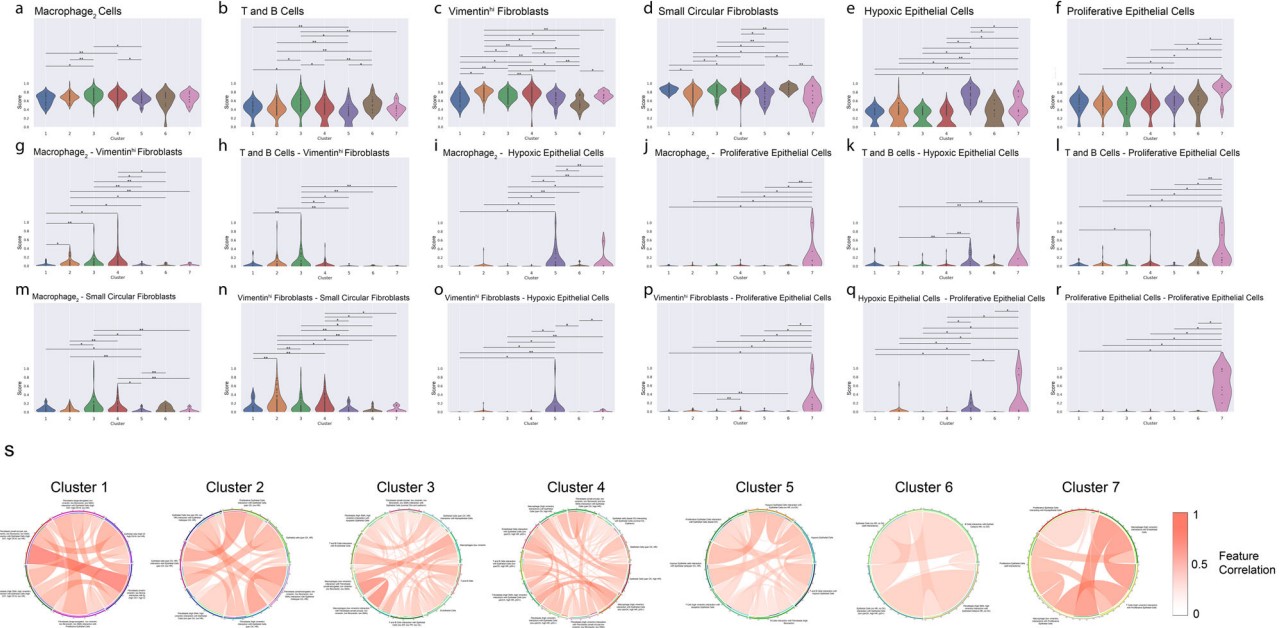

**Fig. 4 Characterization of the single image-based survival subtypes. a–r** Scoring and profiling for the seven survival subtypes based on various cellular phenotypes and cell-cell interaction features. Cell counts and interactions were normalized between 0 and 1 to make comparison possible on the same scale. **a** Macrophage$_2$ Cells, **b** T and B Cells, **c** Vimentin$^{hi}$ Fibroblasts, **d** Small Circular Fibroblasts, **e** Hypoxic Epithelial Cells, **f** Proliferative Epithelial Cells, **g** Macrophage$_2$–Vimentin$^{hi}$ Fibroblasts, **h** T and B Cells–Vimentin$^{hi}$ Fibroblasts, **i** Macrophage$_2$–Hypoxic Epithelial Cells, **j** Macrophage$_2$–Proliferative Epithelial Cells, **k** T and B Cells–Hypoxic Epithelial Cells, **l** T and B Cells–Proliferative Epithelial Cells, **m** Macrophage$_2$–Circular Fibroblasts, **n** Vimentin$^{hi}$ Fibroblasts–Small Circular Fibroblasts, **o** Vimentin$^{hi}$ Fibroblasts–Hypoxic Epithelial Cells, **p** Vimentin$^{hi}$ Fibroblasts–Proliferative Epithelial Cells, **q** Hypoxic Epithelial Cells–Proliferative Epithelial Cells, **r** Proliferative Epithelial Cells–Proliferative Epithelial Cells. Statistical testing was done for $n = 259$ patients using the Mann-Whitney U Test with Benjamini-Hochberg-based false discovery rate (FDR) adjustment. The significant pairs are marked as follows: *: p-value < 0.01. **: 0.01 < p-values < 0.05. Each violin is drawn using the kernel density estimate of the underlying distribution. **s** Circos plots demonstrate the correlation between feature pairs associated with each subpopulation.

scores in interactions between T/B cells and proliferative epithelial cells (Fig. 4l). Subtype 7 stands out with the highest level of proliferative epithelial cells (Fig. 4f) and the second highest level of hypoxic epithelial cells (Fig. 4e). As a result, subtype 7 has the highest scores in the greatest number of TMI-tumor interaction categories (Fig. 4j–l, p–r). In summary, the subtypes 5–7 with some of the worst survival outcomes are highly enriched with hypoxic (subtype 5) or proliferative epithelial cells (subtype 6 and 7), as well as the corresponding interactions between immune cells and these epithelial cells; however, they have much lower interactions between fibroblast and immune cells in the tumor microenvironment (Fig. 4g, h, m, n).

Next, we investigated the higher-order of correlation relationship among the cell-cell interactions for each survival subtype. We calculated the correlations among all the CP, TMI and TCI features and studied the pairs of cell-cell interaction with correlations greater than 0.5 (Fig. 4s). For subtype 1, a high correlation exists between epithelial cell-small, elongated fibroblast interaction and epithelial cell-small circular fibroblast interaction, indicating that they share similar modes of interactions. Similarly, subtype 3 shows a high correlation between macrophage-small elongated fibroblast interaction and macrophage-small circular fibroblast interaction. In subtype 7, proliferative epithelial-macrophage/T-Cell (vimentin-expressing) interaction is highly correlated with interactions between vimentin-expressing macrophage and endothelial cells, further confirming the detrimental and synergistic effect of certain immune cells on the proliferative tumor.

**Discovery of subpopulations of TNBC and luminal A subtypes with atypical survival outcomes**. To uncover new insights into

this survival subtyping, we compared the classification results using molecular subtyping vs. the survival subtyping approach (Fig. 5a). Each molecular subtype is split into multiple survival subtypes, demonstrating the widespread heterogeneity in terms of survival. We dichotomized the 7 survival subgroups into good survival (subgroups 1-4) and bad survival (subgroups 5-7) groups and noticed that for luminal A and TNBC cancers, these subgroups show significantly different survival outcomes (log-ranked p-value < 0.05 for $n = 44$ independent TNBC patients and $n = 166$ independent luminal A patients) from their counterparts (Fig. 5b, e). Despite being a molecular subtype regarded as having the worst prognosis, TNBC actually has a large proportion (61.36%) of atypical good survival patients from clusters 1-4, where cluster 3 is the major cluster among the four (66.66%). On the other hand, while the luminal A subtype generally has the best prognosis among the four molecular subtypes, around 11% of the luminal A breast cancers still belong to the relatively bad survival group. We next investigated the signatures of these atypical subpopulations by differential expression analysis. The atypical good survival subpopulation in TNBC has over-expression of *GATA3* (Fig. 5b, Supplementary Data 5). On the other hand, the atypical bad survival subpopulation in the luminal A subtype has an over-expression of *KRT7* and *ACTA2* and an under-expression of *CDH1* (Fig. 5e, Supplementary Data 6).

We next tested if such atypical subpopulations can be validated in general. We used the UNION-COM (Cao, Bai, Hong, & Wan, 2020) method to perform label-transfer learning based on the pseudo-bulk protein expression in this single-cell dataset and applied the model to the transcripts of the same genes in TCGA-BRCA and METABRIC data. The results show that TCGA-BRCA and METABRIC datasets indeed have such good survival subpopulations in TNBC (Fig. 5c, d)

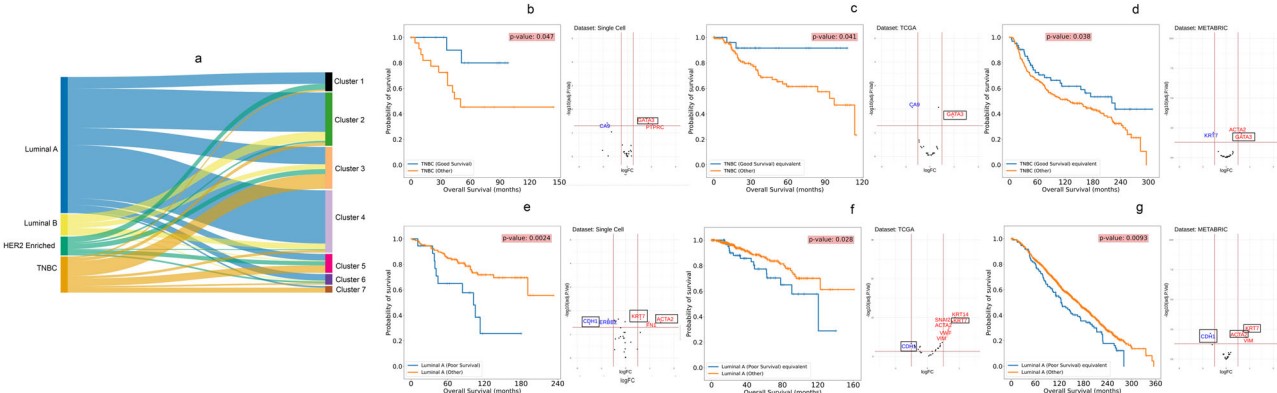

**Fig. 5 Discovery and validation of atypical subpopulations in triple-negative breast cancer (TNBC) and luminal A patients. a** Sankey plot showing the distribution of patients in the seven subpopulations vs. clinical subtypes. **b** Kaplan-Meier (KM) survival plot for the two TNBC subpopulations in the single-cell dataset and the differentially expressed protein biomarkers for the two subpopulations. **c** Validation of the two TNBC subpopulations in TCGA breast cancer data corresponding to those in (**b**) by KM plot and differentially expressed genes. **d** Validation of the two TNBC subpopulations in METABRIC breast cancer data corresponding to those in (**b**) by KM plot and differentially expressed genes. **e** KM survival plot for the two luminal A subpopulations in the single-cell dataset and the differentially expressed protein biomarkers for the two subpopulations. **f** Validation of the two luminal A subpopulations in TCGA breast cancer data corresponding to those in (**e**) by KM plot and differentially expressed genes. **g** Validation of the two luminal A subpopulations in METABRIC breast cancer data corresponding to those in (**e**) by KM plot and differentially expressed genes. (For each subplot, the blue curve represents the subpopulation in context).

and bad survival subpopulations in luminal A patients (Fig. 5f, g). The atypical subpopulations in both validation cohorts have significantly different survival curves (log-rank *p*-value < 0.05 for *n* = 159 independent TCGA-BRCA TNBC samples, *n* = 599 independent TCGA-BRCA luminal A patients, *n* = 299 independent Metabric TNBC patients and *n* = 1369 independent Metabric luminal A patients) compared to their counterparts. Moreover, *GATA3* is also consistently overexpressed in the atypical good survival subpopulations in both the TCGA-BRCA and METABRIC datasets (Fig. 5c, d, Supplementary Data 7 and 8). Additionally, *CA9* shows a common under-expression in the TCGA-BRCA dataset (Fig. 5b, c). Similarly, the matching poor survival luminal A subpopulations in TCGA-BRCA and METABRIC datasets show the same patterns of over-expression of *KRT7* and *ACTA2* and under-expression of *CDH1* (Fig. 5f, g, Supplementary Data 9 and 10). Loss of *CDH1* is a characteristic of invasive lobular breast cancer[33]. Confirming this, we found that lobular type increases from 20.72 to 50.7%, comparing the good vs. bad survival subgroups in TCGA luminal A breast cancers. It also increases from 8.34 to 12.7% for the good vs bad survival subgroups in the luminal A cancers in the METABRIC dataset.

TNBC was previously reported to have six molecular subtypes[29], including 2 basal-like (BL1 and BL2), an immuno-modulatory (IM), a mesenchymal (M), a mesenchymal stem–like (MSL), and a luminal androgen receptor (LAR) subtype, as well as an extra unstable (UNS) subtype. We further cross-checked the survival-based TNBC subtypes here with those reported earlier (Supplementary Fig. 2). We selected six markers (*MKI67, MYC, EGFR, CDH1, SNAI2, TWIST1*) that are in common with the Supplementary Fig. 6 of the original study and our single-cell imaging dataset and relied on the pixels in this heatmap as the normalized gene expression values for the comparative study. We scaled the marker gene intensities from the single-cell imaging data by the RGB values of the pixels. The good TNBC survival group of the image data shows a lower expression in the proliferation markers and myoepithelial markers and a mixed expression in EMT markers (Supplementary Fig. 2b). And the bad TNBC survival group of the image data shows a combination of low and high expressions in Proliferation markers and EMT markers (Supplementary Fig. 2c). From the clustering result of averaged gene expression in two types of subtype systems, the

good TNBC survival subgroup is the closest to MSL and LAR subgroups, whereas the bad TNBC survival group is closest to BL2 and M subgroups (Supplementary Fig. 2d).

## Discussion

Breast cancer is a highly heterogeneous disease, and molecular subtypes based on ER, PR and HER2 statuses are currently the mainstream classification system. In this study, we leverage the strengths of detailed single-cell pathology images and neural-network-based prognosis modeling and define a class of survival-related subtypes for breast cancer. We argue that cellular phenotypes and their interactions provide additional valuable information to predict survival, leading to improved clinical impacts.

The uniqueness of this study lies in several aspects. From the analytical aspect, we explicitly computed hundreds of features describing cell-cell interactions (TMI and TCI) on single-cell imaging data. We used these features as inputs for a neural-network-based survival prediction method called 2-stage Cox-nnet, which highly accurately fits patient survival and is much better than clinical data-based Cox-PH regression. From the biomedical aspect, we have defined seven survival subtypes with distinct profiles of epithelial, immune, and fibroblast cells, as well as their interaction patterns. These survival subtypes amend the molecular subtypes originally based on tumor cell signatures, as each of the molecular subtypes is highly heterogeneous in terms of the patient's prognosis. It also extends beyond previous work that classified breast cancers by immuno-subtypes[34] by considering additionally the interactions among tumor, immune cells, etc.

These survival subpopulations are well characterized by protein markers and cellular phenotypes. The relatively good survival subtypes 1–4 are enriched with CK or HR expression epithelial cells as well as interactions between fibroblasts and other immune cell types, which are lacking in the poorer survival subtypes 5–7. The best survival subtype 1 has a high expression level of luminal CK. Corresponding to this observation, CKs were reported to be associated with better overall survival in breast cancer before[35]. Subtype 2, which also has good survival, has high levels of CK8/18 and HR, which was also reported to be associated with good overall survival[36]. Subtype 5 has a dominant hypoxia phenotype,

demonstrated by the high presence of hypoxia epithelial cells. Hypoxia is associated with resistance against therapies and poor outcomes[37,38]. Despite most of the patients being luminal A subtype, patients in this subgroup show the lowest level of Vimentin[hi] Fibroblasts cells (high vimentin expression, low smooth muscle actin (SMA) expression, and low fibronectin expression)[39]. It also has the 2nd highest level of proliferative epithelial cells, both of which may contribute to poor survival. Subtype 7 has the highest level of proliferative epithelial cells marked by KI-67 expression, associated with poor survival[40,41]. It also has some of the strongest interactions with cells in EMT, suggesting the highest degree of tumor infiltration by immune cells[42]. The highly active TMI observed in hypoxic subtype 5 and proliferative subtype 7 are quite interesting. Subtype 5 has lower proportions of immune cells (e.g., macrophages) but the strongest interactions between macrophages and the hypoxic epithelial cells, the signature epithelial cells of this subtype. It also has the highest interaction between Vimentin[hi] fibroblasts and hypoxic epithelial cells. The apparent paradox between low macrophage concentration vs high interaction with epithelial cells may be explained by the inflammatory cytokines produced by macrophages at the hypoxic site, which decreases tumoricidal activity[43]. Subtype 7 has the highest proportion of proliferative epithelial cells, and 75% of the patients are TNBC. This subtype has the strongest interactions between various cells in the tumor microenvironment (vimentin expression T/B cells, fibroblasts, and macrophages2) and proliferative epithelial cells, as well as interactions between vimentin-expressing macrophages and hypoxic cells. A high degree of immune infiltration was reported in TNBC patients with exhausted T Cell populations[44], which is consistent with our results. This subtype also shows the highest interaction between hypoxic epithelial cells and proliferative epithelial cells. The hypoxic condition may be a result of the increased aggressiveness of proliferating epithelial cells[45], consistent with prior observation of a close relationship between proliferation-hypoxia in other solid cancers such as squamous cell carcinoma[46].

The subtyping system enabled the discovery of atypical survival subpopulations. A good survival subpopulation among TNBC patients (mostly from survival subtype 3) is identified with over-expression of GATA3; and a small population of bad survival subtype among luminal A patients are found, with over-expression of KRT7 and ACTA2 but under-expression of CDH1. These results are robustly verified in TCGA-BRCA and METABRIC datasets. GATA3 is a zinc finger transcription factor involved in cell type differentiation and proliferation. Corresponding to our observations, higher levels of GATA3 were also shown to be correlated with better survival in breast cancer patients[47]. KRT7 encodes cytokeratin-7 and ACTA2 encodes smooth muscle alpha-2 actin, both are cytoskeleton components. KRT7 plays a role in cell migration and epithelial-mesenchymal transition (EMT) pathways and is associated with poor survival subpopulations for ovarian cancer[48,49]. Similarly, ACTA2 was reported to be a marker for poor prognosis in lung cancer[50]. CDH1 is a transmembrane glycoprotein that is primarily responsible for cell adhesion, and its downregulation in tumors led to increased invasiveness in breast cancer and lung cancer[51]. Thus, the work here paved the foundation for improving the subtyping of TNBC and luminal A cancers using the above-mentioned biomarkers. Moreover, therapeutics targeting these molecules may be effective at improving TNBC and luminal A cancer patients' survival.

We further investigated the marker genes that have high correlations (Pearson's correlation coefficient >0.5 or <−0.5) with the cell-cell interactions, using the two extreme subtypes 1 and 7 (Supplementary Fig. 3). Many above-described relationships

with regard to prognosis are confirmed again. For subgroup 1 (Supplementary Fig. 3a), there exist positive correlations between CDH1 and epithelial-epithelial cell interactions. Since the loss of CDH1 is a characteristic of invasive lobular breast cancer[33], a strong positive correlation between CDH1 and the epithelial-epithelial interactions does indicate good survival. On the other hand, positive correlations of MKI67 with epithelial-epithelial interaction and T-cell-epithelial cell interaction in group 7 (Supplementary Fig. 3b) indicate poor survival, as expected[52]. Interestingly, GATA3 is negatively correlated with T-cell-epithelial cell interaction in group 7, a cell-cell interaction signature for poor survival. However, it is positively correlated with epithelial-epithelial interaction in group 1 with good survival.

While it is true that the cost and time of the multiplex imaging platform currently do not allow live-time assistance to clinicians with precision treatment, this technology is highly reproducible and futuristic. It has already been used for the diagnosis of other diseases like leukemia[53]. This technology has provided information on disseminated tumor cell phenotypes that frequently deviate from the clinical disease subtype[54]. It also advanced our understanding of the intra-tumor heterogeneity and how the interactions among different cell types in tumors and tumor microenvironments affect patient outcomes[15,16,55]. It is expected that in the near future, with the reduction of cost and time and assistance of AI tools and user-friendly computational tools, such technologies will eventually be adopted by pathology and be more informative to clinicians.

In summary, the imaging cytometry data at the single-cell resolution has enabled an unprecedented opportunity to explicitly study cells in the tumor and tumor microenvironment, as well as their interactions. By taking advantage of CP, TMI and TCI features, novel survival subtypes are identified in breast cancers, each with distinct profiles and more molecular and survival homogeneity. Moreover, this subtyping system allows us to identify good survival subpopulations in TNBC and bad survival subpopulations in luminal A cancers, with robust biomarkers in multiple population cohorts. The work lays down the foundation to integrate single-cell resolution information for survival prediction at a large population scale. It has the potential to improve breast cancer prognosis prediction and patient-stratified treatment in the future.

## Data availability

The breast cancer imaging data supporting this study (image mass cytometry data, phenograph neighborhood information, clinical features and patient prognosis outcome) are available online at Zenodo per the original study (https://doi.org/10.5281/zenodo.3518284)[56]. The source data for generating the main and supplementary figures are available at the Zenodo repository (https://doi.org/10.5281/zenodo.10038601)[57]. The data for validation of the results are available at the TCGA portal (https://www.cancer.gov/ccg/research/genome-sequencing/tcga) for TCGA-TNBC data and at cbioportal (https://www.cbioportal.org/study/summary?id=brca_metabric) for METABRIC data.

## Code availability

The source code to generate the figures and feature datasets for this work is available at https://github.com/lanagarmire/BC_imaging[57].

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

## Acknowledgements

We thank Dr. Hartland Jackson for providing data for this project. We thank Yuheng Du in the Garmire group for valuable discussions. This research was supported by grants R01 LM012373 and LM012907 awarded by NLM and R01 HD084633 awarded by NICHD to L.X.G.

## Author contributions

L.X.G. envisioned this project and supervised the study. S.Y. performed the initial data analysis, generated the main figures, and wrote the manuscript. S.Z. characterized the atypical molecular subtypes, revised the manuscript and figures, tested the code used in this study and made it available on the Github repository. B.H. helped with subgroup selection and interpretation. Y.D. helped test the atypical molecular subtypes with regard to the cell proportions.

## Competing interests

The authors declare no competing interests.
