## [Corrected peer review file · Communications Medicine]

Reviewers' comments:

Reviewer #1 (Remarks to the Author):

The paper describes a method to train a CNN based on cell-phenotype features and cell-cell interaction features. Main technical innovation is the integration of a survivalmodel-aware CNN architecture in the system, enabling the uncovering of several clinically relevant (survival related) subtypes of survival. This leads to a much better understanding of which features may be predictive for poor or better survival.

Strong point of the paper is that it finds subtypes in the Hart Jackson paper data that were not uncovered in the original paper. The detail subgrouping gives more detail in the discriminating features and their association with survival.

Strong point is also that the derived discriminative markers also generalize in two independent datasets: TCGA-BRCA and METABRIC.

I have little to no corrections to the paper, not do I have severe concerns. In fact, I'm quite enthusiastic about the paper: it's a solid piece of work, which includes method innovation: integration of cell-phenotype, cell-cell interaction and some patient specific survival metric into a unified CNN design is the way to go. My only recommendation would be to rewrite a bit the section on how the cell-cell specific features are computed. How do you define the concept of when two cells are considered neighbors? This section is a bit foggy at the moment and could be more clearly described.

Reviewer #2 (Remarks to the Author):

Comment

This paper presents a new breast cancer survival subtype system by using single-cell imaging data. The authors extract features describing cell-cell interactions and cellular phenotypes from a large cohort of breast cancer patients, then apply a neural-network-based survival prediction model to predict patient survival. Although the authors provide some insights into bridging single-cell level information towards population-level survival prediction, its guidance for clinical practice in breast cancer is limited. In a word, this study requires some revisions. Some comments to consider:

1. In this study, imaging mass cytometry data from 259 patients were used to construct seven subtypes related to the survival of breast cancer patients using the COX-nnet model. However, there are limitations in the clinical application of these subtypes for two reasons. Firstly, these subtypes do not take into account the three important molecular subtypes that are widely recognized in breast cancer. Secondly, due to cost and time constraints, the sequencing technology used in the database is not suitable for assisting clinicians in precision treatment.

2. Breast cancer is a highly heterogeneous disease, and the molecular subtypes based on ER, PR, and HER2 status are currently the mainstream classification system. The molecular classification of breast cancer has evolved to six subtypes for TNBC[1]. This study proposes some TNBC patients belonging to the subtype 3 with better prognosis. Therefore, it is important for the author to specify which subtype these patients correspond to within the six subtypes of TNBC. Additionally, it would be beneficial if the authors could provide an explanation as to why these particular TNBC patients are grouped together with other luminal and HER2-positive patients.

3. Tumor cells impact patient survival through two main factors: proliferation and metastasis. However, in this study, the Cox-PH model used as the baseline model only considered clinical

information including ER status, PR status, HER2 status, and tumor grade. It fails to include variables that represent metastatic characteristics. The AJCC Cancer Staging Manual (8th edition) assesses breast cancer based on tumor diameters (T), regional lymph node metastases (N), and distant metastases (M). Therefore, it is highly inappropriate for this baseline model to exclude the TNM staging system which evaluates disease progression in breast cancer. The authors should incorporate additional clinical variables into the baseline model to ensure its accuracy.

4. In this study, the author proposes a total of 405 features, consisting of 378 cell-cell interaction features and 27 cellular phenotypes. However, only 400 features are included in the survival model. Please provide an explanation for the discrepancy in the number of features.

5. This study investigates the interactions between cells in the tumor microenvironment of breast cancer at a single-cell spatial level. However, its clinical application is limited. It is suggested that the authors further explore specific subtype's microenvironmental characteristics to uncover potential mechanisms of tumor microenvironment interaction that could impact overall breast cancer.

[1] Lehmann, B. D., Bauer, J. A., Chen, X., Sanders, M. E., Chakravarthy, A. B., Shyr, Y., & Pietersen, J. A. (2011). Identification of human triple-negative breast cancer subtypes and preclinical models for selection of targeted therapies. *The Journal of clinical investigation*, 121(7), 2750–2767.

Addressing the reviewers' comments:

Reviewer #1

I have little to no corrections to the paper, nor do I have severe concerns. In fact, I'm quite enthusiastic about the paper: it's a solid piece of work, which includes method innovation: integration of cell-phenotype, cell-cell interaction and some patient specific survival metric into a unified CNN design is the way to go. My only recommendation would be to rewrite a bit the section on how the cell-cell specific features are computed. How do you define the concept of when two cells are considered neighbors? This section is a bit foggy at the moment and could be more clearly described.

Thank you for your positive affirmation. Much appreciated!! We have revised the cell-cell interaction feature section. The previous work had identified 27 cellular phenotypes that described the histopathological landscape of breast cancer. We used the same concept as them to define direct neighbor cells: those are within 4 pixels (4 μ m) distance of the target cell.

Reviewer #2

1. In this study, imaging mass cytometry data from 259 patients were used to construct seven subtypes related to the survival of breast cancer patients using the COX-nnet model. However, there are limitations in the clinical application of these subtypes for two reasons. Firstly, these subtypes do not take into account the three important molecular subtypes that are widely recognized in breast cancer. Secondly, due to cost and time constraints, the sequencing technology used in the database is not suitable for assisting clinicians in precision treatment.

Thanks for the comments. We added these aspects in the "Discussion" section. The current molecular subtypes are used as part of the baseline model to predict patient survival. We also draw a lot of comparisons between the single-cell imaging based classification vs. those from the molecular subtypes throughout the report (eg. Fig. 2, Fig. 3d and 3e, Fig. 5). While it is true that the cost and time of the multiplex imaging platform currently do not allow live-time assistance to clinicians to precision treatment yet, this technology nevertheless significantly advanced our understanding of the intra-tumor heterogeneity and how the interactions among different cell types affect patient outcome, as demonstrated by others (Ali et al., 2020; Danenberget al., 2022; Jackson et al., 2020) and us here.

Image cytometry is a futuristic technology and has already been established in the diagnosis of other diseases like leukemia (Grimwade et al., 2017). It is expected that in the near future, with the reduction of cost and time and assistance of AI tools and user-friendly computational tools, such technologies will eventually be adopted by pathology and be more informative to clinicians.

References:

Ali, H. R., Jackson, H. W., Zanotelli, V. R. T., Danenberg, E., Fischer, J. R., Bardwell, H., Provenzano, E., CRUK IMAXT Grand Challenge Team, Rueda, O. M., Chin, S.-F., Aparicio, S., Caldas, C., & Bodenmiller, B. (2020). Imaging mass cytometry and multiplatform genomics define the phenogenomic landscape of breast cancer. *Nature Cancer*, *1*(2), 163–175. <https://doi.org/10.1038/s43018-020-0026-6>

Danenberg, E., Bardwell, H., Zanotelli, V. R. T., Provenzano, E., Chin, S.-F., Rueda, O. M., Green, A., Rakha, E., Aparicio, S., Ellis, I. O., Bodenmiller, B., Caldas, C., & Ali, H. R. (2022). Breast tumor microenvironment structures are associated with genomic features and clinical outcome. *Nature Genetics*, *54*(5), 660–669. <https://doi.org/10.1038/s41588-022-01041-y>

Jackson, H. W., Fischer, J. R., Zanotelli, V. R. T., Ali, H. R., Mechera, R., Soysal, S. D., Moch, H., Muenst, S., Varga, Z., Weber, W. P., & Bodenmiller, B. (2020). The single-cell pathology landscape of breast cancer. *Nature*, *578*(7796), 615–620. <https://doi.org/10.1038/s41586-019-1876-x>

Grimwade, L. F., Fuller, K. A., & Erber, W. N. (2017). Applications of imaging flow cytometry in the diagnostic assessment of acute leukaemia. *Methods*, *112*, 39–45. <https://doi.org/10.1016/j.ymeth.2016.06.023>

2. Breast cancer is a highly heterogeneous disease, and the molecular subtypes based on ER, PR, and HER2 status are currently the mainstream classification system. The molecular classification of breast cancer has evolved to six subtypes for TNBC[1]. This study proposes some TNBC patients belonging to subtype 3 with better prognosis. Therefore, it is important for the author to specify which subtype these patients correspond to within the six subtypes of TNBC. Additionally, it would be beneficial if the authors could provide an explanation as to why these particular TNBC patients are grouped together with other luminal and HER2-positive patients.

[1] Lehmann, B. D., Bauer, J. A., Chen, X., Sanders, M. E., Chakravarthy, A. B., Shyr, Y., & Pietersen, J. A. (2011). Identification of human triple-negative breast cancer subtypes and preclinical models for selection of targeted therapies. *The Journal of clinical investigation*, *121*(7), 2750–2767.

Thank you for the suggestion. The referred reference proposing seven subtypes of TNBC does not provide gene expression data unfortunately, neither could we get the authors to respond to our request. Still we wish to answer the reviewer's comment even though we could not obtain the original gene expression data. For this goal, we selected six markers (MKI67, MYC, EGFR, CDH1, SNAI2, TWIST1) that are in common in the original study (their Fig S6) and our single-cell imaging dataset, and relied on the pixels in their heatmap as the normalized gene expression values for the comparative study.

We made a new supplementary Fig S2, with the following subfigures: (a) the seven molecular subgroups of TNBC in the reference, based on the common markers (b) hierarchical clustering of the bad-survival TNBC subtype in the single-cell image data; (c) hierarchical clustering of the good-survival TNBC subtype in the single-cell image data; (d) the heatmap on averaged expression of subtypes defined by us and the six subtypes defined in the previous study.

The good TNBC survival group of the image data shows a lower expression in the proliferation markers and myoepithelial markers and a mixed expression in EMT markers (**Figure S2b**). And the bad TNBC survival group of the image data shows a combination of low and high expressions in Proliferation markers and EMT markers (**Figure S2c**). From the clustering result of averaged gene expression in two types of subtype systems, the good TNBC survival subgroup is the closest to MSL and LAR subgroups, whereas the bad TNBC survival group is closest to BL2 and M subgroups(**Figure S2d**).

3. Tumor cells impact patient survival through two main factors: proliferation and metastasis. However, in this study, the Cox-PH model used as the baseline model only considered clinical information including ER status, PR status, HER2 status, and tumor grade. It fails to include variables that represent metastatic characteristics. The AJCC Cancer Staging Manual (8th edition) assesses breast cancer based on tumor diameters (T), regional lymph node metastases (N), and distant metastases (M). Therefore, it is highly inappropriate for this baseline model to exclude the TNM staging system which evaluates disease progression in breast cancer. The authors should incorporate additional clinical variables into the baseline model to ensure its accuracy.

Thank you for the valuable suggestion. We now have included the TNM staging system as part of the baseline model, in addition to the other features considered earlier (ER

status, PR status, HER2 status, and tumor grade). We focused on comparing the new baseline model vs the two-stage Cox-nnet models in the revised Fig 2.

4. In this study, the author proposes a total of 405 features, consisting of 378 cell-cell interaction features and 27 cellular phenotypes. However, only 400 features are included in the survival model. Please provide an explanation for the discrepancy in the number of features.

Thanks for the question. Among the 405 feature pairs, 1 immune-stromal feature (macrophage interacting with Large Elongated Fibroblasts), 2 immune-epithelial features (macrophage interacting with Hypoxic Epithelial Cells and macrophage interacting with CK7⁺CK^{hi}Cadherin^{hi} Epithelial Cells), 1 stromal-stromal feature (Large Elongated Fibroblasts self interaction) and 1 stromal-epithelial (Large Elongated Fibroblasts interacting with Myoepithelial cells) feature were dropped since these pairwise phenotype-phenotype interactions calculation returned 0 (lack of neighbors within the immediate vicinity), hence the final total number of features was 400. We revised the manuscript to be more clear.

5. This study investigates the interactions between cells in the tumor microenvironment of breast cancer at a single-cell spatial level. However, its clinical application is limited. It is suggested that the authors further explore specific subtype's microenvironmental characteristics to uncover potential mechanisms of tumor microenvironment interaction that could impact overall breast cancer.

Thanks. We showed the interactions between tumor microenvironment cell types in the original Figure 4. Given the comments of this reviewer, we further investigated the molecular mechanism of the survival subgroup 1 and 7 (**Figure S3**), by examining the correlations between the molecular marker genes and the signature cell-cell interactions shown in Fig. 4S. As described before, the most important TME cells interacting with epithelial cell pairs in subgroup 1 are fibroblasts, whereas, the most important TME cells interacting with epithelial cell pairs in subgroup 7 are macrophages (high vimentin) and T cells.

Many previously described relationships with regards to prognosis and marker gene expression are confirmed again. For subgroup 1 (**Figure S3a**), there exist positive correlations between CDH1 and epithelial-epithelial cell interactions. Since the loss of CDH1 is a characteristic of invasive lobular breast cancer (Dossus et al, 2015), a strong positive correlation between CDH1 and the epithelial-epithelial interactions does

indicate good survival. On the other hand, positive correlations of MKI67 with epithelial-epithelial interaction and T-cell-epithelial cell interaction in group 7 (**Figure S3b**) indicates poor survival, as expected (Soliman et al., 2016). Interestingly, GATA3 is negatively correlated with T-cell-epithelial cell interaction in group 7, a cell-cell interaction signature for poor survival. However, it is positively correlated with epithelial-epithelial interaction in group 1 with good survival.

References:

Dossus, L., & Benusiglio, P. R. (2015). Lobular breast cancer: incidence and genetic and non-genetic risk factors. *Breast Cancer Research: BCR*, 17, 37. <https://doi.org/10.1186/s13058-015-0546-7>

Soliman, N. A., & Yussif, S. M. (2016). Ki-67 as a prognostic marker according to breast cancer molecular subtype. *Cancer Biology & Medicine*, 13(4), 496–504. <https://doi.org/10.20892/j.issn.2095-3941.2016.0066>

REVIEWERS' COMMENTS:

Reviewer #1 (Remarks to the Author):

The authors have addressed my previous comments in a satisfactory manner. I therefore believe the paper can now be accepted as-is.

Reviewer #2 (Remarks to the Author):

The author answers my questions very well, and I don't think this study needs further revision. This study is suitable for publication in this journal.